# Female *Psammomys obesus* Are Protected from Circadian Disruption-Induced Glucose Intolerance, Cardiac Fibrosis and Adipocyte Dysfunction

**DOI:** 10.3390/ijms25137265

**Published:** 2024-07-01

**Authors:** Joanne T. M. Tan, Cate V. Cheney, Nicole E. S. Bamhare, Tasnim Hossin, Carmel Bilu, Lauren Sandeman, Victoria A. Nankivell, Emma L. Solly, Noga Kronfeld-Schor, Christina A. Bursill

**Affiliations:** 1Vascular Research Centre, Lifelong Health Theme, South Australian Health and Medical Research Institute, Adelaide, SA 5000, Australia; cate.cheney@sahmri.com (C.V.C.); nicole.bamhare@outlook.com (N.E.S.B.); tasnim.hossin@outlook.com (T.H.); lauren.sandeman@sahmri.com (L.S.); victoria.nankivell@sahmri.com (V.A.N.); emma.solly@sahmri.com (E.L.S.); 2Adelaide Medical School, Faculty of Health and Medical Sciences, University of Adelaide, Adelaide, SA 5005, Australia; 3School of Zoology, Tel Aviv University, Tel Aviv 69978, Israel; carmel.bilu@gmail.com (C.B.); nogaks@tauex.tau.ac.il (N.K.-S.)

**Keywords:** inflammation, cellular hypertrophy, adipocyte differentiation, browning

## Abstract

Circadian disruption increases the development of cardiovascular disease and diabetes. We found that circadian disruption causes glucose intolerance, cardiac fibrosis and adipocyte tissue dysfunction in male sand rats, *Psammomys obesus*. Whether these effects occur in female *P. obesus* is unknown. Male and female *P. obesus* were fed a high energy diet and exposed to a neutral (12 light:12 dark, control) or short (5 light:19 dark, circadian disruption) photoperiod for 20 weeks. Circadian disruption impaired glucose tolerance in males but not females. It also increased cardiac perivascular fibrosis and cardiac expression of inflammatory marker *Ccl2* in males, with no effect in females. Females had reduced proapoptotic *Bax* mRNA and cardiac *Myh7:Myh6* hypertrophy ratio. Cardiac protection in females occurred despite reductions in the clock gene *Per2*. Circadian disruption increased adipocyte hypertrophy in both males and females. This was concomitant with a reduction in adipocyte differentiation markers *Pparg* and *Cebpa* in males and females, respectively. Circadian disruption increased visceral adipose expression of inflammatory mediators *Ccl2*, *Tgfb1* and *Cd68* and reduced browning marker *Ucp1* in males. However, these changes were not observed in females. Collectively, our study show that sex differentially influences the effects of circadian disruption on glucose tolerance, cardiac function and adipose tissue dysfunction.

## 1. Introduction

There is a well-established link between cardiovascular disease (CVD) and type 2 diabetes mellitus (T2DM) with co-morbidity associated with an elevated risk of adverse outcomes [1]. Individuals with diabetes or pre-diabetes are at a higher risk of developing cardiovascular disease than their non-diabetic counterparts, with cardiovascular complications being the major cause of mortality and morbidity in diabetic patients [2,3]. Diabetes causes a large array of myocardial abnormalities that can result in heart failure including left ventricular hypertrophy, impaired tolerance to ischemic injury and pathophysiological changes in the myocardium including increased inflammation and fibrosis that can cause diabetic cardiomyopathy [2]. Concurrently, variations in fat distribution, adipocyte tissue dysfunction and adiposity-associated inflammation and adipose tissue dysfunction are strongly implicated in the development of T2DM and atherosclerotic cardiovascular disease [4,5]. Furthermore, visceral adipose tissue is known to be more metabolically active and less insulin sensitive than subcutaneous adipose tissue [6].

Disruption to the circadian system has been recently implicated in increasing the development and progression of cardiovascular disease and T2DM [7]. Circadian rhythms occur within a 24 h period and are either indirectly or directly controlled by circadian clocks present in almost every cell [8]. The circadian system is a major regulator of human metabolism, regulating gene expression, release of hormones, body temperature, activity patterns and energy expenditure [7]. Disruption of circadian rhythms therefore has a significant impact on these major metabolic systems that they control.

Biological sex plays a crucial role in the progression of CVD and T2DM. Globally, the prevalence of cardiovascular disease and diabetes is greater in men than women [9]. Emerging research has identified differences in the circadian system based on gender/biological sex. Male–female differences in circadian regulation occur at the individual brain cell level through many factors; sex chromosome differences, ion channels, differential circulating hormones and receptor expression specialization all play a role in distinguishing circadian differences by sex [10]. However, research into the impact of the circadian system on cardiovascular disease and diabetic complications are currently majorly conducted in single sex (mainly male) studies.

*Psammomys obesus* (*P. obesus*) are a unique polygenic diurnal rodent model that, when placed on a high energy diet, spontaneously develop human-like T2DM and obesity. This has been shown to cause severe alterations of cardiac structure and activation of inflammatory processes [11]. Furthermore, we have recently shown that circadian disruption by short exposure in combination with a high energy diet impairs glucose tolerance, increases cardiac fibrosis and drives adipocyte dysfunction [3,12,13]. In this study, we sought to determine if sex influences the effects of circadian disruption on glucose tolerance, cardiac fibrosis and adipocyte dysfunction using male and female *P. obesus*.

We found that males exposed to a high-energy diet and short photoperiod had impaired glucose tolerance, consistent with our previous findings [14]. This was associated with increased perivascular cardiac fibrosis and elevations in cardiac expression of the inflammatory cytokine *Ccl2*. In contrast, females exposed to a short photoperiod did not have impaired glucose tolerance. Gene analyses also showed a reduction in cardiac expression of the proapoptotic marker *Bax* and the cardiac hypertrophy marker, the *Myh7:Myh6* ratio. The short photoperiod significantly increased adipocyte hypertrophy in both males and females. However, we observed significant sex differences in gene expression of markers reflective of adipose tissue dysfunction. Males exposed to a high-energy diet and short photoperiod had increased visceral adipose expression of inflammatory mediators *Ccl2*, *Tgfb1* and *Cd68* concomitant with a reduction in browning marker *Ucp1*. In contrast, these changes were not observed in females. Collectively, our study shows that the effects of circadian disruption on glucose tolerance, cardiac function and adipose tissue dysfunction differ between sexes.

## 2. Results

### 2.1. Short Photoperiod Impairs Glucose Tolerance in Males but Not Females

Short photoperiod exposure did not affect body weights within males and females when compared to their respective neutral photoperiod counterparts (Table 1). However, female *P. obesus* were significantly lighter than the males within the same photoperiod group. All animals gained weight throughout the duration of the study although the weight change is varied within the groups (Appendix A). No differences were observed in baseline glucose levels prior to the oral glucose tolerance test (Table 1). Blood glucose levels of males exposed to a short photoperiod were significantly higher (SP-M: 16.5 ± 8.2 mM, Figure 1) compared to their respective neutral photoperiod counterparts (NP-M: 8.4 ± 4.3 mM, *p* < 0.05). However, blood glucose levels were not significantly different between females exposed to either the neutral or short photoperiod (NP-F: 7.1 ± 1.7 mM vs. SP-F: 11.3 ± 3.5 mM, *p* = 0.1844).

### 2.2. Short Photoperiod Induced Cardiac Perivascular Fibrosis in Male P. obesus

We previously demonstrated that circadian disruption in male *P. obesus* induces myocardial perivascular fibrosis [3]. We next compared male and female myocardial perivascular fibrosis in sections of the left ventricle by staining for the presence of collagen around the heart vessels. A significant increase in collagen deposition was observed around the myocardial vessels of male *P. obesus* exposed to a short photoperiod (1.44 ± 0.91% collagen/vessel area, Figure 2), compared to males exposed to a neutral photoperiod (0.64 ± 0.41% collagen/vessel area, *p* < 0.05). In contrast, there was a significant decrease in collagen deposition in female *P. obesus* exposed to the short photoperiod compared to their respective male counterparts (SP-F: 0.67 ± 0.43 vs. SP-M: 1.44 ± 0.91, *p* < 0.05).

### 2.3. Differential Gene Changes with Short Photoperiod Exposure in Males and Females

We next sought to determine if there were any differential gene changes following the short photoperiod exposure. Increased myocardial inflammation is a central cause of perivascular fibrosis in T2DM. Assessment of changes in the mRNA levels of inflammatory chemokine *Ccl2* revealed that female *P. obesus* exposed to the short photoperiod had significantly lower cardiac *Ccl2* (88.3 ± 77.7%, Figure 3a) compared to male *P. obesus* exposed to the short photoperiod (241.2 ± 142.9%, *p* < 0.05). Additional pairwise analyses within each biological sex revealed that male *P. obesus* exposed to the short photoperiod had significantly elevated cardiac *Ccl2* (241.2 ± 142.9%, Figure 3b), compared to neutral photoperiod counterparts (100.0 ± 75.5%, *p* < 0.05). Myocyte apoptosis occurs as a consequence of myocardial inflammation and fibrosis and is a precursor to heart failure. No differences were observed in *Bax* mRNA levels across all four groups irrespective of sex or photoperiod exposure (Figure 3c). Interestingly, pairwise analyses showed that females exposed to the short photoperiod had significantly reduced *Bax* levels when compared to neutral-photoperiod females (SP-F: 43.4 ± 40.4% vs. NP-F: 143.6 ± 99.3%, *p* < 0.05, Figure 3d). Finally, we measured the expression of *Per2*, a key clock gene that indicates circadian rhythmicity of the molecular clock machinery present in every cell [15]. *Per2* expression shows a daily rhythm and for these studies, animals were euthanized at ZT2. At this timepoint, *Per2* was not different across all four groups (Figure 3e). However, pairwise analyses showed that *Per2* was significantly reduced with short photoperiod exposure in females when compared to their respective neutral photoperiod counterparts (SP-F: 74.7 ± 36.6% vs. NP-F: 195.1 ± 124.7%, *p* < 0.05, Figure 3f). Expression heatmaps and XY plots of each individual animal showed differential expression within groups (Appendix A). Logistic regression was used to analyze the relationship between cardiac *Ccl2*, *Bax*, and *Per2* with either of the two variables, biological sex and photoperiod (Table 2). It was found that, holding all other predictor variables constant, cardiac *Ccl2* and *Bax* expression is dependent on the photoperiod. We also observed some correlation with cardiac *Myh7:Myh6* with sex although this did not quite reach statistical significance (*p* = 0.058).

### 2.4. Differential Effects of a Short Photoperiod on Cardiac Hypertrophy Gene Expression and Cardiomyocyte Size in Male and Female P. obesus

We next measured for changes in cardiac hypertrophy markers myosin heavy chain (MHC)α (adult isoform) and MHCβ (fetal isoform), represented by genes *Myh6* and *Myh7*, respectively. No differences were observed in the ratio of *Myh7:Myh6* across all four groups (Figure 4a). However, direct comparison of short photoperiod animals alone showed that the ratio of *Myh7:Myh6* was lower in females compared to males, reaching near significance (SP-F: 62.9 ± 36.2% vs. SP-M: 100.0 ± 30.3%, *p* = 0.0521, Figure 4b). This reduction, however, did not translate into an observable phenotypic change in cardiomyocyte size (Figure 4c). Expression heatmaps and XY plots of each individual animal showed differential *Myh7:Myh6* expression within groups (Appendix A). Logistic regression analyses showed that *Myh7:Myh6* mRNA is independent of biological sex and photoperiod (Table 2).

### 2.5. Short Photoperiod Induces Adipocyte Hypertrophy in Visceral and Subcutaneous Depots

Increased adipocyte size has been reported to be associated with reduced insulin sensitivity and T2DM [16,17]. Both males and females exposed to a short photoperiod had a significant increase in area size in visceral adipocytes (SP-M: 5953 ± 1982 µm^2^ and SP-F: 6097 ± 1801 µm^2^, Figure 5) compared to their respective neutral photoperiod counterparts (NP-M: 3042 ± 798 µm^2^ and SP-F: 3494 ± 769 µm^2^, *p* < 0.05 for both). Interestingly, short photoperiod only increased subcutaneous adipocyte size in females (SP-F: 7044 ± 1701 µm^2^ vs. NP-F: 3968 ± 2379 µm^2^, *p* < 0.05, Figure 6).

### 2.6. Short Photoperiod Reduces Visceral Expression of Adipocyte Differentiation Markers

Adipocyte differentiation is a functional protector against insulin resistance of adipocytes and adipocyte hypertrophy. PPARy and C/EBPα are critical transcription factors involved in adipogenesis [18]. There was a significant reduction in visceral *Pparg* expression in males exposed to a short photoperiod compared to their neutral photoperiod counterparts (SP-M: 38.8 ± 17.9% vs. NP-M: 100.0 ± 53.6%, *p* < 0.05) but no differences were observed in females (Figure 7a). Subcutaneous *Pparg* expression was not significantly different across the groups (Figure 7b). In females, short photoperiod exposure led to lower visceral *Cebpa* mRNA levels compared to neutral-photoperiod females (SP-F: 63.1 ± 33.1% vs. NP-F: 123.5 ± 43.2%, *p* < 0.05, Figure 7c). However, male visceral *Cebpa* was not different. No significant differences were observed in subcutaneous *Cebpa* expression (Figure 7d). Expression heatmaps and XY plots of each individual animal showed differential expression within groups (Appendix A). Logistic regression was used to analyze the relationship between adipocyte differentiation markers *Pparg* and *Cebpa* with either of the two variables, biological sex and photoperiod (Table 3). It was found that, holding all other predictor variables constant, visceral *Pparg* is dependent on the photoperiod.

### 2.7. Short Photoperiod Reduces Expression of Browning Marker Ucp1 in Males but Not Females

Recent studies have reported that white adipocytes can undergo browning to form a more thermogenic fat phenotype that oxidizes glucose and lipids via UCP1-mediated thermogenesis [19]. Browning has been shown to improve insulin sensitivity and protect against metabolic dysfunction in mice, posing it as a potential therapeutic approach to treat T2DM [20]. We next measured the expression of *Ucp1* to determine the degree of browning. There was a significant reduction in both visceral and subcutaneous *Ucp1* expression in males exposed to a short photoperiod (Figure 8a,b). In contrast, SP-F have significantly elevated *Ucp1* mRNA levels in both adipose depots when compared to SP-M. Expression heatmaps and XY plots showed differential expression within individual animals in each group (Appendix A). Logistic regression found that visceral and subcutaneous *Ucp1* expression is dependent on sex (Table 3).

### 2.8. Short Photoperiod Induces Visceral Inflammation in Males but Not Females

Adipose tissue inflammation is linked to the development of T2DM [21]. Inflammatory cytokines such as CCL2 and TGF-β1 are secreted by adipocytes in response to excessive lipid accumulation. The induction of an inflammatory environment promotes the infiltration and expansion of macrophages to the adipose site, which disrupts the homeostasis of insulin signaling in adipose tissue [21]. In order to examine the effect of the photoperiod on inflammation among males and females, mRNA levels of *Ccl2*, macrophage marker *Cd68* and *Tgfb1* were measured. Short photoperiod exposure increased visceral *Ccl2* levels by three-fold in males compared to their neutral photoperiod controls (SP-M: 323.1 ± 137.2% vs. NP-M: 100.0 ± 98.8%, *p* < 0.001, Figure 9a). However, this induction was not seen in females. Comparison of both short photoperiods showed that visceral *Ccl2* levels were significantly lower in females compared to males (SP-F: 92.2 ± 26.6% vs. SP-M: 323.1 ± 137.2%, *p* < 0.001). In contrast, the short photoperiod significantly reduced subcutaneous *Ccl2* levels in males (SP-M: 14.1 ± 7.6% vs. NP-M: 100.0 ± 77.3%, *p* < 0.05, Figure 9b). Neutral-photoperiod females had significantly reduced *Ccl2* levels (18.6 ± 14.9%) compared to their male counterparts (*p* < 0.05). Short-photoperiod females had a five-fold increase in *Ccl2* (103.1 ± 59.3%) compared to neutral-photoperiod females (*p* < 0.05). The short photoperiod significantly increased *Tgfb1* expression in males (*p* < 0.05) and this induction was protected in females (*p* < 0.01, Figure 9c) with no differences seen in subcutaneous *Tgfb1* (Figure 9d). Similarly, the short photoperiod caused a three-fold increase in visceral *Cd68* levels in males (SP-M: 287.1 ± 156.9% vs. NP-M: 100.0 ± 37.7%, *p* < 0.05, Figure 9e). This increase was not seen in females, with short-photoperiod females having significantly lower *Cd68* levels (116.0 ± 74.8%, *p* < 0.05). No differences were observed in subcutaneous *Cd68* expression (Figure 9f). Expression heatmaps and XY plots showed differential expression of adipocyte inflammatory genes within individual animals in each group (Appendix A). Logistic regression was used to analyze the relationship between adipocyte inflammatory markers *Ccl2*, *Tgfb1*, *Cd68* with either of the two variables, biological sex and photoperiod (Table 4). It was found that, holding all other predictor variables constant, visceral *Ccl2* expression is dependent on sex.

## 3. Discussion

Circadian disruption increases a plethora of metabolic-related complications including cardiovascular disease and diabetes [7]. Using the unique rodent model of *P. obesus* we investigated the role of sex on the effects of circadian disruption on the onset of cardiovascular disease and diabetes. We found distinct differences between males and females following 20 weeks of short photoperiod (5 h light:19 h dark) exposure. Exposure to a short photoperiod in males (SP-M) caused glucose intolerance, increased cardiac fibrosis and elevated myocardial levels of inflammatory chemokine *Ccl2*. By contrast, these changes did not occur in females. Furthermore, we found that female *P. obesus* hearts expressed lower levels of the proapoptotic marker *Bax* and the cardiac hypertrophy marker, the *Myh7:Myh6* ratio, but this was not translated into increased cardiomyocyte size histologically. While short photoperiod exposure increased adipocyte hypertrophy in both males and females, females were protected against the effects of circadian disruption on adiposity-associated inflammation and impaired browning.

Circadian rhythms have a key role in the regulation of glucose and insulin in the body, with large-scale control regulated by the suprachiasmatic nucleus (SCN) [22]. As the master biological clock, the SCN regulates daily rhythms in response to light signals. Glucose uptake has been demonstrated to be impaired in rodents with lesions in their SCN, highlighting the importance of light signal processing in metabolism [22]. In the current study, we have shown that when exposed to circadian disruption for 20 weeks, male *P. obesus* demonstrate features of T2DM development such as an increased glucose intolerance. This finding is consistent with previous studies that show circadian disruption acutely impacts glycemic control and increases the risk for transition to diabetes [23]. This effect did not, however, occur in females who demonstrated no significant change in their glucose tolerance following circadian disruption. These findings are consistent with a recent study that showed that circadian disruption impairs glucose homeostasis in male but not in female mice and is dependent on gonadal sex hormones [14,24].

Hyperglycemia increases cardiac fibrosis through induction of pro-inflammatory signalling leading to increased collagen deposition [25]. The organization of this newly formed collagen into a fibrotic scar occurs interstitially or around myocardial vessels (perivascular fibrosis) as measured in this study. The increase in perivascular fibrosis observed in short-photoperiod males is consistent with previous studies [3]. Males maintained on a high-energy diet and a short photoperiod displayed an increase in perivascular fibrosis compared to low-energy short-photoperiod males and high-energy neutral-photoperiod males [3]. The absence of a significant increase in collagen deposition observed in the short-photoperiod females (SP-F) may be indicative of protective mechanisms that delay or prevent cardiomyocyte dysfunction and impaired extracellular matrix production. This protective effect against fibrosis observed in females may be strongly attributed to hormonal differences between sexes. Recent research into female protection against cardiac fibrosis has demonstrated that estrogen has a protective effect against harmful extracellular matrix remodelling and ensuing cardiac fibrosis through alteration of fibroblast proliferation [26].

We also observed sex-specific changes in key regulators that drive cardiac function. Firstly, we observed a significant increase in elevations in myocardial *Ccl2* mRNA levels in males exposed to a short photoperiod. CCL2 is reported to be a pivotal mechanistic driver of hyperglycemic-driven and ischemic myocardial fibrosis, in which it is secreted from activated macrophages [27]. This provides a mechanism to explain the higher levels of cardiac fibrosis in males. The induction of *Ccl2* is likely to have been induced by the circadian disruption-induced glucose intolerance seen in males. This may also explain why females did not have increased myocardial *Ccl2*, as they did not develop glucose intolerance following circadian disruption. Consistent with this, hyperglycemia is reported to induce inflammation, specifically CCL2, in cardiomyocytes [28]. We observed a significant reduction in the levels of proapoptotic marker *Bax*. Bax upregulation, a marker of intrinsic apoptosis, is implicated under conditions of pathophysiological states associated with ischemic myocardium or increased inflammation and stretching/hypertrophy of myocardial cells [29]. Intrinsically regulated cardiomyocyte apoptosis is considered an essential process in the progression to heart failure; however, estrogen is implicated to suppress activation of intrinsic apoptotic pathways, a mechanism of female protection [30,31]. Lower levels of *Bax* in the short-photoperiod females may be indicative of a protective response to decrease cardiomyocyte apoptosis.

There was a significant reduction in the cardiac hypertrophy marker ratio *Myh7:Myh6* in females. Despite this, no changes in cardiomyocyte hypertrophy were noted histologically. This finding may indicate that changes in hypertrophy genes precede physical changes in cardiomyocyte size, as reported previously [3]. These findings show evidence that shift workers experiencing circadian disruption may have an elevated risk of developing T2DM and cardiac fibrosis that causes heart failure, which is more pronounced in males than females. *Per2* is a key clock gene that indicates circadian rhythmicity of the molecular clock machinery present in every cell [3]. Here, we observed a significant reduction of *Per2* in SP-F at ZT2, with a non-significant increase in SP-M. Our previous study showed that cardiac *Per2* peaked with a short photoperiod at ZT8 in males [3]. This decrease in *Per2* in females, despite displaying protection against glucose intolerance and cardiac fibrosis contradicts previous findings that demonstrate deletion of *Per2* induces severe cardiac consequences in males [32]. The protective effect demonstrated in females may be attributed to the actions of female sex hormone estrogen as an overarching mechanism of protection in females exposed to circadian disruption, even under conditions of disturbed circadian rhythmicity [33]. We have previously reported that *Per2* expression shows a daily rhythm across several tissues [3,12,13]. The reduction in females could be indicative of disturbed rhythmicity as a result of circadian disruption or that the peak time may have shifted.

We next explored the effects of sex on circadian disruption of adipose tissue dysfunction. Circadian disruption significantly increased the visceral adipocyte size in both males and females. Adipocyte hypertrophy is a strong indicator of metabolic impairment, to which numerous studies have reported that an increased adipocyte size is associated with the progression of T2DM [34]. Visceral adipose tissue is more metabolically active than subcutaneous adipose tissue and is related to metabolic health and overall insulin resistance [35]. Body fat distribution is known to be significantly different between males and females and is a strong predictor of disease risk [36,37]. Interestingly, circadian disruption only increased subcutaneous adipocyte size in female *P. obesus*. Women have more subcutaneous adipose tissue, while men predominantly have visceral adipose fat tissue distributed around the abdominal organs [38,39,40]. Nevertheless, women can also have an upper-body obese phenotype; the reduced metabolic disease risk in women has been attributed to the predisposition to store body fat in the lower body region [41,42]. Despite this significant increase in adipocyte hypertrophy, circadian disruption did not affect glucose tolerance in females. This suggests that females are protected from the effects of disrupted circadian rhythm and adipocyte hypertrophy on glucose intolerance.

Concomitant with increased visceral adipocyte size, we found that circadian disruption had differential effects in visceral expression of these adipogenesis drivers, with reduced visceral *Pparg* expression in males while visceral *Cebpa* levels were lower in females. PPARγ and C/EBPα play a significant role in adipocyte differentiation [43,44,45]. A previous study showed that 60% of adipocyte differentiation genes bind to both PPARγ and C/EBPα simultaneously [45]. Hormones are crucial regulators of adipose tissue and are critical for adipocyte development and function. Estrogen plays a significant role in the regulation of adipose tissue development in both males and females [46]. Women tend to have increased sensitivity compared to men, as seen in past studies where adipocytes of female mice had an increased lipogenic capacity compared to adipocytes from males [47]. These studies highlight sex-specific differences in the regulation of adipocyte differentiation.

Over recent years, brown adipose tissue (BAT) and the browning of white adipose tissue (WAT) have become excellent therapeutic targets to treat T2DM. Brown adipocytes have the ability to accelerate substrate oxidation, enabling an increased substrate delivery and energy turnover [19]. The stimulation of WAT creates a new type of thermogenic brown adipocyte that promotes energy turnover and an increased metabolic activity by oxidizing glucose and lipids. Our study found that circadian disruption reduced the expression of the browning marker *Ucp1* in both male visceral and subcutaneous adipose tissue. These findings contradict previous studies that show that short (winter) photoperiod exposure usually leads to increased UCP1 expression and increased non-shivering thermogenesis capacity [48,49,50]. Our previous study reported variations in *Ucp1* rhythmicity in both adipose depots [13]. It is likely that the timing of *Ucp1* peak may not have been captured in this study. Interestingly, *Ucp1* expression was not affected in females. Females tend to have a higher thermogenic capacity of BAT compared to males [51,52]. Estrogen has the ability to enhance thermogenesis in BAT and promotes the browning of WAT [46]. Despite the increase in adipocyte hypertrophy seen in both depots, increased browning of female adipose tissue may be the link that protects them against glucose intolerance.

Adipose tissue inflammation is a crucial factor for the pathogenesis of obesity-linked insulin resistance and T2DM. Obesity is considered a state of chronic low-grade inflammation that is characterized by elevated circulating cytokines and chemokines [20]. Excessive lipid accumulation drives the secretion of inflammatory cytokines CCL2 and TGF-β1, which promote the accumulation of macrophages [53]. Studies have reported depot-specific differences in inflammatory function, with increased expression of inflammatory markers and macrophage infiltration in visceral adipose tissue compared to subcutaneous adipose tissue [54]. Previous studies have shown that a high energy diet results in weight gain that is accompanied by the infiltration of inflammatory cells such as macrophages [4]. CCL2 is produced by adipocytes and is associated with adiposity [55], with circulating CCL2 levels being elevated in human obese subjects and is associated with obesity-related parameters [56]. Additionally, TGF-β1 has been shown to promote macrophage accumulation, collagen deposition and drive the remodeling of fat tissue in obese mice [57]. We found that circadian disruption strikingly increased visceral *Ccl2*, *Cd68* and *Tgfb1* expression in males. The rise of such markers in visceral depots of short-photoperiod males support the finding that visceral adipose tissue is suggestive of a chronic inflammatory state. Importantly, this induction in visceral inflammation was not seen in *P. obesus* females, suggesting that females are protected from circadian disruption-induced visceral adipose inflammation. Interestingly, we found that circadian disruption elevated subcutaneous *Ccl2* levels in females, which may contribute to the increase in subcutaneous adipocyte size seen.

Multiple logistic regression analyses showed that visceral *Ucp1* and *Ccl2*, and subcutaneous *Ucp1* gene expression was dependent on sex while cardiac *Ccl2* and *Bax*, and visceral *Pparg* gene expression was dependent on photoperiod. Expression heatmaps and XY plots showed differential gene expression across individual animals within each group, suggesting varied responses. It is likely that a larger sample population would provide further insights into the influence of biological sex and photoperiod. Unfortunately, while we were able to identify significant effects of biological sex on these key inflammatory and adipocyte markers at a transcriptional level, we were unable to validate these findings at a protein level. This is due to the limited availability of antibodies to robustly detect these markers in *Psammomys obesus*.

## 4. Materials and Methods

### 4.1. Animal Studies

All experimental procedures followed the NIH guidelines for the care and use of laboratory animals and were approved by the Institutional Animal Care and Use Committee (IACUC) of Tel Aviv University (Permit Number: L15055). This study was conducted on HsdHu diabetes-prone male and female 6-month-old sand rats (*Psammomys obesus*). All animals were maintained on a low-energy diet (Koffolk Ltd., Tel Aviv, Israel) prior to the start of the experiment, and monitored for body weight and glucose levels. Animals were then allocated to experimental groups according to the weights and blood glucose levels to avoid baseline bias. Animals were fed *ad libitum* the standard rodent diet (21% protein, 4% fat, 4% Crude fiber; Product No.: 2018; Koffolk Ltd.), which contains an established higher caloric density that facilitates the development of type 2 diabetes mellitus in *P. obesus* [58]. Adult male and female sand rats (n = 6–9/group) were exposed to either a neutral (NP, 12 h light:12 h dark) or short (SP, 5 h light:19 h dark) photoperiod. This photoperiod regimen is based off the activity pattern exhibited by *P. obesus* in the wild where they are active for approximately 5 h from midday during the winter [58]. *P. obesus* were acclimated to their allocated photoperiod and maintained in these conditions for 20 weeks before the start of manipulation and testing. Light illuminance was set at 800 lux, wavelength 420–780 nm (5834 K). For these studies, the animals were divided across four groups: 1) males exposed to NP (NP-M), 2) males exposed to SP (SP-M), 3) females exposed to NP (NP-F) and 4) females exposed to SP (SP-F).

At week 20, animals underwent an oral glucose tolerance test. Animals received a bolus of glucose at 2 mg/kg body weight, 2 h after lights on (ZT 2; ZT = Zeitgaber Time; ZT 0 = the time of lights on). Blood glucose levels were measured from tail vein nicks at baseline and 120 min later. At the conclusion of the study, animals were euthanized at ZT2, and blood was collected. The heart, visceral and subcutaneous adipose tissues were collected and snap-frozen for further analysis.

### 4.2. Histological Analysis

To assess the extent of cardiac fibrosis, a portion of the right ventricle of the heart was formalin fixed and paraffin embedded and then sectioned on a microtome (5 µm). Two sections were stained with Masson’s Trichome (Sigma, St. Louis, MO, USA). Two sections per animal were selected and photographed at 400× total magnification. Heart vessels were selected at random. Each vessel was then photographed under × 400 magnification, with five photos per section across two sections (n = 10 images per animal). Vessel area was determined by tracing around the outside of the vessel using a polygon measure tool. A spatial calibration macro was then used to define the area of collagen (stained in blue) surrounding the vessel. Perivascular cardiac fibrosis was determined through calculation of a ratio for the area of collagen surrounding the vessels divided by the vessel area [3]. All images were taken using an AxioLab microscope attached to a camera (Zeiss, Oberkochen, Baden-Württemberg, Germany) and then, analyzed with Image-Pro Premier 9.2 (64 bit).

To determine the magnitude of cardiomyocyte hypertrophy, two sections were stained with Hematoxylin and Eosin (Thermofisher, Waltham, MA, USA) and then, imaged to define cell morphology. Analysis was focused on regions of cardiomyocytes with centralized nuclei [3]. The cytoplasm area (stained in pink) was measured with a fixed threshold setting. The smart segmentation (nuclei count) tool was used to count the number of hematoxylin-stained nuclei per view, excluding objects of < 2 μm^2^. Determination of cardiomyocyte size was calculated by dividing the area of the cytoplasm by the number of nuclei [3].

To determine adipocyte size, 5 µm visceral and subcutaneous adipose tissue sections were stained with Hematoxylin and Eosin (Thermofisher, Waltham, MA, USA) to define adipocyte morphology. Two sections per animal were selected and two fields of view were imaged per section with an AxioLab microscope attached to a camera (Zeiss, Oberkochen, Baden-Württemberg, Germany) at × 400 total magnification. To analyze adipocyte cell size, two sections were analyzed with two fields of view/section randomly selected as far away from the other where possible. Analyses at × 20 magnification allows the measurement of between 500–1000 adipocytes per animal. The area of the adipocytes was determined by manually tracing the interior of all the adipocytes within the field of view using ZEN lite 2.3 and the average adipocyte area per animal was determined [13].

### 4.3. Gene expression Analysis

Total RNA was extracted from the apex region of the myocardium, visceral and subcutaneous adipose tissues with TRI^®^ reagent (Sigma-Aldrich, St. Louis, MO, USA). RNA concentration and quality were assessed spectrophotometrically. An amount of 500 ng of total RNA was converted to cDNA using the iScript cDNA Synthesis Kit (Bio-Rad, Hercules, CA, USA). Quantitative real-time PCR was performed to assess (1) cardiac expression of *Myh6*, *Myh7*, *Ccl2*, *Bax*, *Per2* and *Cyclophilin* and (2) adipose expression of *Pparg*, *Cebpa*, *Ucp1*, *Ccl2*, *Cd68*, *Tgfb1* and *Cyclophilin* using previously published primers [3,11,13,59]. Relative gene expression was calculated using the ^ΔΔ^*Ct* method, normalized to *Cyclophilin* and respective NP groups.

### 4.4. Statistics

Data is presented as mean ± SD. Normal distribution of data was determined using the Shapiro–Wilk test. All analyses were performed unblinded. Differences between groups were calculated using either one-way ANOVA (Šídák’s multiple comparison test post hoc), Kruskal–Wallis test (Dunn’s multiple comparison test post hoc), unpaired *t*-test or Mann–Whitney test, where appropriate. Multiple logistic regression analyses were performed to determine the relationship between cardiac, visceral adipose and subcutaneous adipose gene expression with either of the two variables, biological sex and photoperiod. Significance was set at a two-sided *p* < 0.05.

## 5. Conclusions

Overall, we have shown, that circadian disruption induces impaired glucose tolerance, cardiac fibrosis and adipocyte dysfunction in male *P. obesus*. Importantly, female *P. obesus* are protected from these detrimental effects of circadian disruption. These findings provide further understanding into the role of sex on the effects of circadian disruption. This study has implications for correcting circadian disruption as using sex-specific interventions to reduce the risk of diabetes and cardiovascular disease.

## Figures and Tables

**Figure 1 ijms-25-07265-f001:**
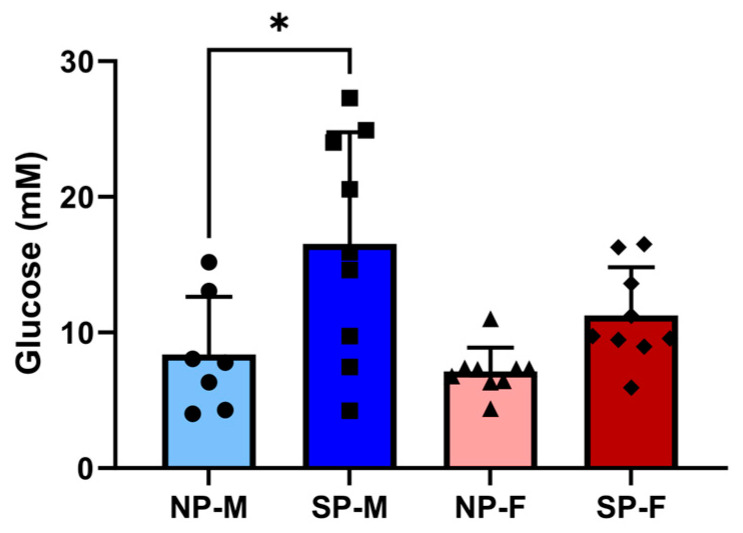
Short photoperiod impairs glucose tolerance in males but not females. Male and female *P. obesus* (n = 6–9/group) were exposed to neutral (12 h light:12 h dark) or short (SP, 5 h light:19 h dark) photoperiods and a high-energy diet for 20 weeks. Oral glucose tolerance tests were performed where animals received a bolus administration of glucose (2 g/kg body weight). Blood glucose levels were measured 120 min post-glucose administration. Data presented as mean ± SD. * *p* < 0.05 by one-way ANOVA (Šídák’s post hoc comparison). NP-M: neutral-photoperiod males (circles), SP-M: short-photoperiod males (squares), NP-F: neutral-photoperiod females (triangles), and SP-F: short-photoperiod females (diamonds).

**Figure 2 ijms-25-07265-f002:**
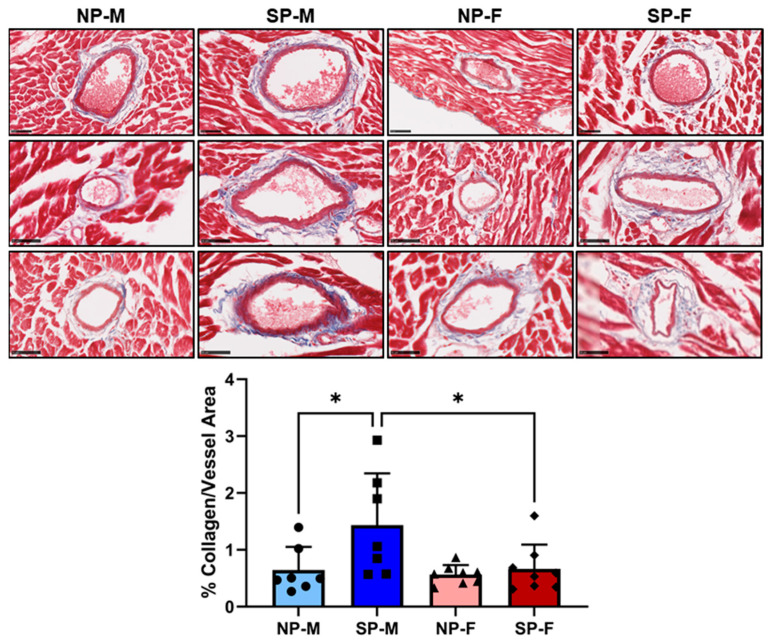
Short photoperiod induced cardiac perivascular fibrosis in male *P. obesus*. Male and female *P. obesus* (n = 6–9/group) were exposed to neutral (12 h light:12 h dark) or short (SP, 5 h light:19 h dark) photoperiods and a high-energy diet for 20 weeks. Representative images of Masson’s trichrome-stained hearts depicting collagen (blue) deposition surrounding the vessels. Perivascular fibrosis was analyzed by determining area of blue staining around selected vessels and normalized to the vessel area. Scale bar: 50 μm. Data presented as mean ± SD. * *p* < 0.05 by one-way ANOVA (Šídák’s post hoc comparison). NP-M: neutral-photoperiod males (circles), SP-M: short-photoperiod males (squares), NP-F: neutral-photoperiod females (triangles), and SP-F: short-photoperiod females (diamonds).

**Figure 3 ijms-25-07265-f003:**
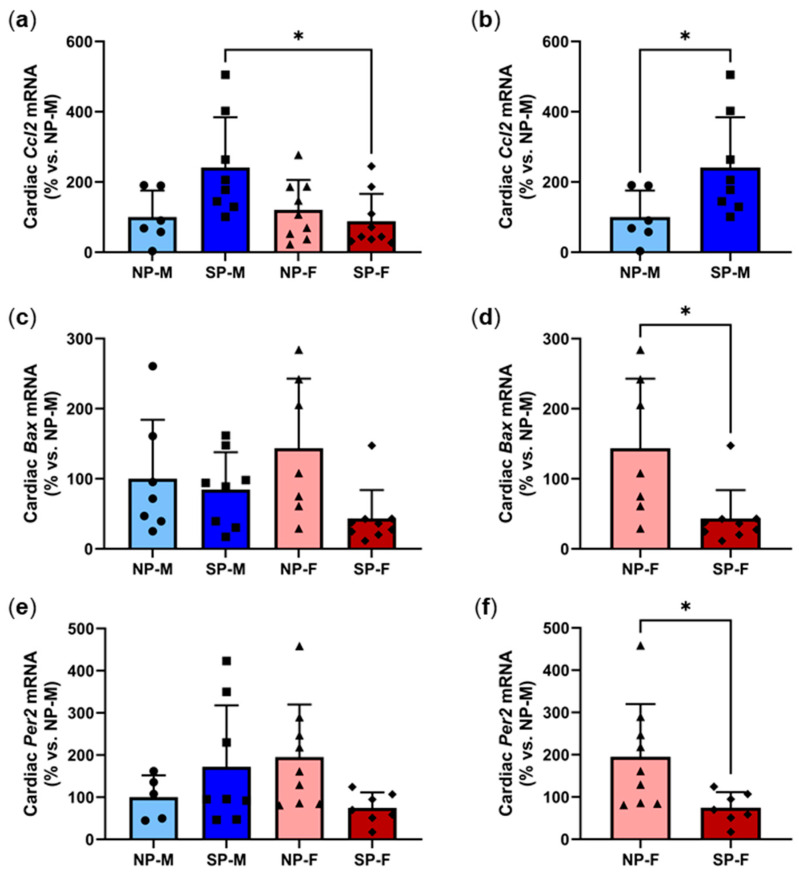
Differential gene changes with short photoperiod exposure in males and females. Male and female *P. obesus* (n = 6–9/group) were exposed to neutral (12 h light:12 h dark) or short (SP, 5 h light:19 h dark) photoperiods and a high-energy diet for 20 weeks. Cardiac expression of *Ccl2* (**a**) for all groups and (**b**) in males alone, *Bax* (**c**) for all groups and (**d**) in females alone and *Per2* (**e**) for all groups and (**f**) in females alone, normalized using the ^ΔΔ^Ct method to *Cyclophilin* and NP-M. Data presented as mean ± SD. * *p* < 0.05 using one-way ANOVA or unpaired Student’s *t*-test. NP-M: neutral-photoperiod males (circles), SP-M: short-photoperiod males (squares), NP-F: neutral-photoperiod females (triangles), and SP-F: short-photoperiod females (diamonds).

**Figure 4 ijms-25-07265-f004:**
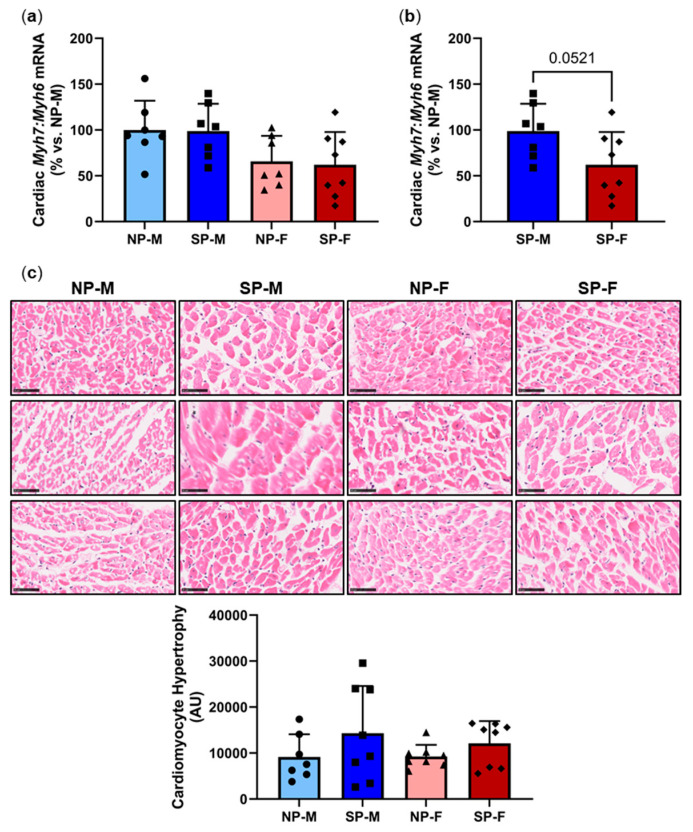
Differential effects of a short photoperiod on cardiac hypertrophy gene expression and cardiomyocyte size in male and female *P. obesus*. Male and female *P. obesus* (n = 6–9/group) were exposed to neutral (12 h light:12 h dark) or short (SP, 5 h light:19 h dark) photoperiods and a high-energy diet for 20 weeks. Cardiac expression of *Myh7:Myh6* mRNA (**a**) for all groups and (**b**) in short photoperiod (SP)-animals alone. (**c**) Representative images of H&E-stained hearts. Myocyte size (area) was analyzed by determining the area of eosin staining divided by the number of nuclei/image. Scale bar: 50 μm. Data presented as mean ± SD. Statistical analysis performed using one-way ANOVA or unpaired Student’s *t*-test. NP-M: neutral-photoperiod males, SP-M: short-photoperiod males, NP-F: neutral-photoperiod females, and SP-F: short-photoperiod females.

**Figure 5 ijms-25-07265-f005:**
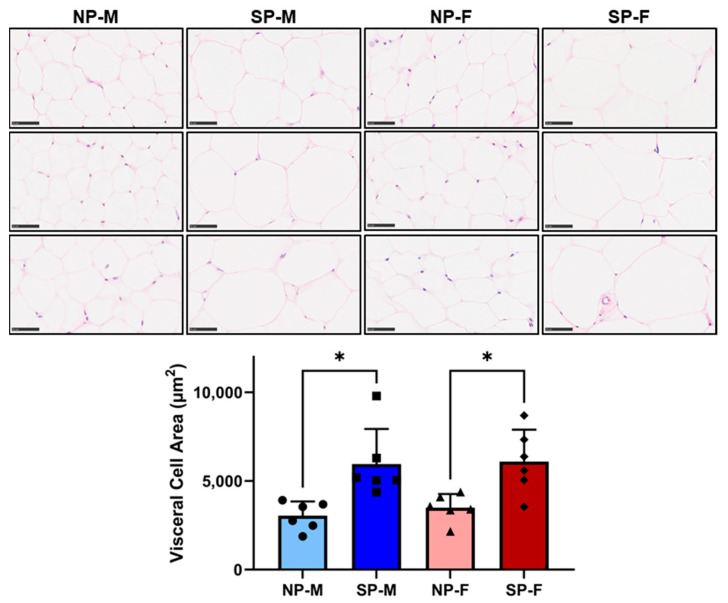
Short photoperiod induces adipocyte hypertrophy in visceral and subcutaneous depots. Male and female *P. obesus* (n = 6/group) were exposed to neutral (12 h light:12 h dark) or short (SP, 5 h light:19 h dark) photoperiods and a high-energy diet for 20 weeks. Representative images and corresponding average visceral cell area calculated from average cell area measurement/animal per group. Scale bars: 50 μm. Data presented as mean ± SD. * *p* < 0.05 by Kruskal–Wallis (Dunn’s *post hoc* comparison). NP-M: neutral-photoperiod males, SP-M: short-photoperiod males, NP-F: neutral-photoperiod females, and SP-F: short-photoperiod females.

**Figure 6 ijms-25-07265-f006:**
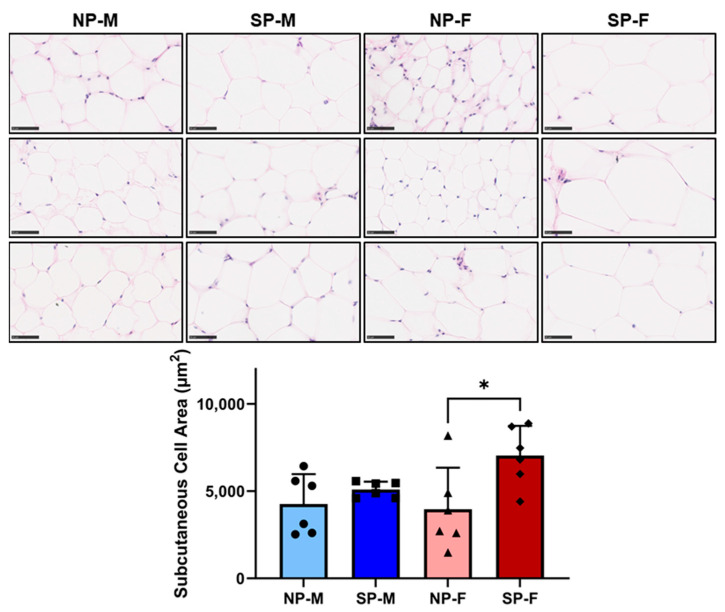
Short photoperiod induces adipocyte hypertrophy in visceral and subcutaneous depots. Male and female *P. obesus* (n = 6/group) were exposed to neutral (12 h light:12 h dark) or short (SP, 5 h light:19 h dark) photoperiods and a high-energy diet for 20 weeks. Representative images and corresponding average subcutaneous cell area calculated from average cell area measurement/animal per group. Scale bars: 50 μm. Data presented as mean ± SD. * *p* < 0.05 by one-way ANOVA (Šídák’s post hoc comparison). NP-M: neutral-photoperiod males, SP-M: short-photoperiod males, NP-F: neutral-photoperiod females, and SP-F: short-photoperiod females.

**Figure 7 ijms-25-07265-f007:**
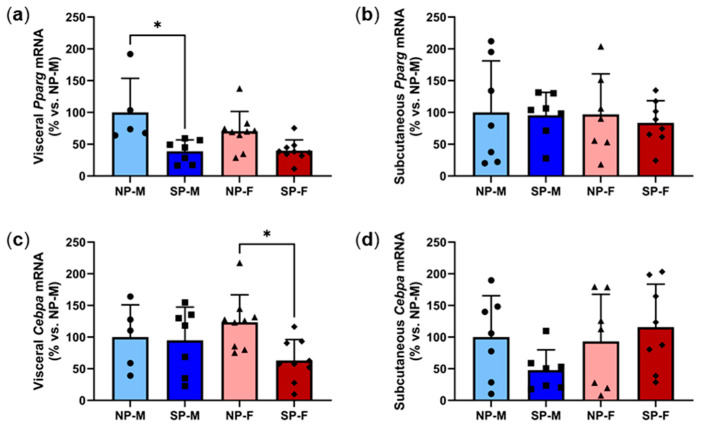
Short photoperiod reduces visceral expression of adipocyte differentiation markers. Male and female *P. obesus* (n = 6–9/group) were exposed to neutral (12 h light:12 h dark) or short (SP, 5 h light:19 h dark) photoperiods and a high-energy diet for 20 weeks. (**a**) Visceral *Pparg*, (**b**) subcutaneous *Pparg*, (**c**) visceral *Cebpa*, and (**d**) subcutaneous *Cebpa*, normalized using the ^ΔΔ^Ct method to *Cyclophilin* and NP-M. Data presented as mean ± SD. * *p* < 0.05 by Kruskal–Wallis (Dunn’s post hoc comparison) or one-way ANOVA (Šídák’s post hoc comparison). NP-M: neutral-photoperiod males, SP-M: short-photoperiod males, NP-F: neutral-photoperiod females, and SP-F: short-photoperiod females.

**Figure 8 ijms-25-07265-f008:**
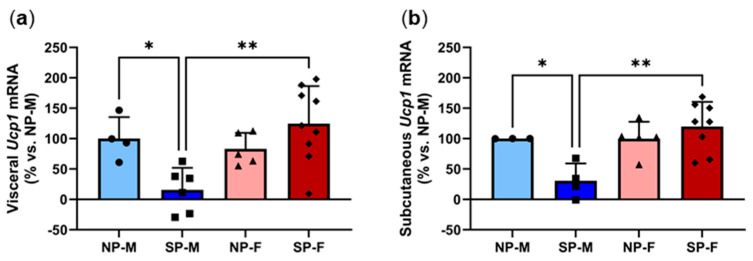
Short photoperiod reduces expression of browning marker *Ucp1* in males but not females. Male and female *P. obesus* (n = 6–9/group) were exposed to neutral (12 h light:12 h dark) or short (SP, 5 h light:19 h dark) photoperiods and a high-energy diet for 20 weeks. (**a**) Visceral and (**b**) subcutaneous *Ucp1* expression, normalized using the ^ΔΔ^Ct method to *Cyclophilin* and NP-M and log transformed. Data presented as mean ± SD. * *p* < 0.05, ** *p* < 0.01 using one-way ANOVA.

**Figure 9 ijms-25-07265-f009:**
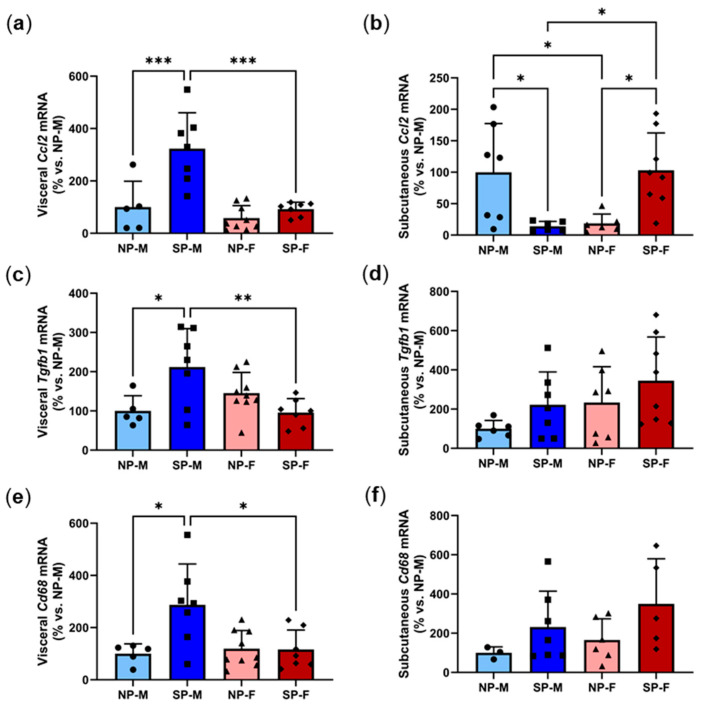
Short photoperiod reduces visceral expression of adipocyte differentiation markers. Male and female *P. obesus* (n = 6–9/group) were exposed to neutral (12 h light:12 h dark) or short (SP, 5 h light:19 h dark) photoperiods and a high-energy diet for 20 weeks. (**a**) Visceral *Ccl2*, (**b**) subcutaneous *Ccl2*, (**c**) visceral *Tgfb1*, (**d**) subcutaneous *Tgfb1*, (**e**) visceral *Cd68*, and (**f**) subcutaneous *Cd68*, normalized using the ^ΔΔ^Ct method to *Cyclophilin* and NP-M. Data presented as mean ± SD. * *p* < 0.05, ** *p* < 0.01, *** *p* < 0.001 by one-way ANOVA (Šídák’s post hoc comparison). NP-M: neutral-photoperiod males, SP-M: short-photoperiod males, NP-F: neutral-photoperiod females, and SP-F: short-photoperiod females.

**Table 1 ijms-25-07265-t001:** Body weights and baseline glucose levels.

	NP-M	SP-M	NP-F	SP-F
Body weight (g)	260.4 ± 13.8	258.1 ± 27.9	217.6 ± 40.0 *	223.1 ± 12.5 ^#^
Baseline glucose (mM)	6.4 ± 3.4	7.2 ± 2.8	6.1 ± 2.1	6.5 ± 2.5

Data presented as mean ± SD. * *p* < 0.05 vs. NP-M, ^#^
*p* < 0.05 vs. SP-M by one-way ANOVA (Šídák’s post hoc comparison). NP-M: neutral photoperiod (NP, 12 h light:12 h dark) males, SP-M: short photoperiod (SP, 5 h light:19 h dark) males, NP-F: neutral-photoperiod females, and SP-F: short-photoperiod females.

**Table 2 ijms-25-07265-t002:** Multiple regression associations of cardiac *Ccl2*, *Bax*, *Per2*, and *Myh7:Myh6* with biological sex and photoperiod.

Independent Variable	Dependent Variable	Estimate	95% CI	*p* Value	*p* Summary
Biological Sex	Intercept	34.56	2.056 to 1976	0.035	*
	*Ccl2*	0.981	0.9531 to 0.9987	0.108	ns
	*Bax*	1.008	0.9726 to 1.047	0.670	ns
	*Per2*	1.002	0.9897 to 1.016	0.752	ns
	*Myh7:Myh6*	0.9712	0.9366 to 0.9977	0.058	ns
Photoperiod	Intercept	4.17	0.2710 to 133.3	0.339	ns
	*Ccl2*	1.026	1.007 to 1.056	0.037	*
	*Bax*	0.9561	0.9092 to 0.9905	0.034	*
	*Per2*	1.001	0.9902 to 1.015	0.831	ns
	*Myh7:Myh6*	0.9877	0.9544 to 1.017	0.424	ns

* *p* < 0.05.

**Table 3 ijms-25-07265-t003:** Multiple regression associations of adipocyte differentiation markers *Pparg* and *Cebpa* and browning marker *Ucp1* with biological sex and photoperiod.

Adipose Tissue Depot	Independent Variable	Dependent Variable	Estimate	95% CI	*p* Value	*p* Summary
Visceral	Biological Sex	Intercept	0.4042	0.01679 to 4.918	0.504	ns
		*Pparg*	0.994	0.9665 to 1.024	0.636	ns
		*Cebpa*	0.9986	0.9761 to 1.021	0.895	ns
		*Ucp1*	1.025	1.006 to 1.056	0.033	*
	Photoperiod	Intercept	330,417	59.25 to 2.548 × 10^15^	0.075	ns
		*Pparg*	0.8176	0.5991 to 0.9374	0.038	*
		*Cebpa*	1.009	0.9337 to 1.112	0.820	ns
		*Ucp1*	0.9891	0.9329 to 1.026	0.603	ns
Subcutaneous	Biological Sex	Intercept	0.0003406	7.166 × 10^−9^ to 0.1203	0.040	*
		*Pparg*	1.054	1.005 to 1.150	0.096	ns
		*Cebpa*	0.9648	0.9034 to 1.004	0.162	ns
		*Ucp1*	1.099	1.029 to 1.249	0.039	*
	Photoperiod	Intercept	1.31	0.05196 to 32.01	0.864	ns
		*Pparg*	1.006	0.9845 to 1.033	0.583	ns
		*Cebpa*	0.9996	0.9785 to 1.021	0.970	ns
		*Ucp1*	0.9966	0.9670 to 1.027	0.814	ns

* *p* < 0.05.

**Table 4 ijms-25-07265-t004:** Multiple regression associations of adipocyte inflammatory markers *Ccl2*, *Tgfb1* and *Cd68* with biological sex and photoperiod.

Adipose Tissue Depot	Independent Variable	Dependent Variable	Estimate	95% CI	*p* Value	*p* Summary
Visceral	Biological Sex	Intercept	17.64	1.585 to 562.8	0.045	*
		*Ccl2*	0.9823	0.9622 to 0.9956	0.035	*
		*Tgfb1*	0.9938	0.9667 to 1.017	0.596	ns
		*Cd68*	1.003	0.9830 to 1.020	0.728	ns
	Photoperiod	Intercept	0.2356	0.01742 to 1.860	0.213	ns
		*Ccl2*	1.011	0.9999 to 1.029	0.106	ns
		*Tgfb1*	0.9894	0.9657 to 1.012	0.351	ns
		*Cd68*	1.011	0.9925 to 1.035	0.296	ns
Subcutaneous	Biological Sex	Intercept	0.1616	0.005404 to 1.471	0.158	ns
		*Ccl2*	1.108	1.009 to 1.318	0.115	ns
		*Tgfb1*	1.011	0.9997 to 1.030	0.137	ns
		*Cd68*	0.9884	0.9626 to 1.001	0.236	ns
	Photoperiod	Intercept	0.3918	0.04577 to 2.635	0.348	ns
		*Ccl2*	1.029	0.9774 to 1.099	0.306	ns
		*Tgfb1*	0.9961	0.9867 to 1.004	0.349	ns
		*Cd68*	1.007	0.9983 to 1.023	0.210	ns

* *p* < 0.05.

## Data Availability

The original contributions presented in the study are included in the article. Further inquiries can be directed to the corresponding author/s.

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
