# Peer review of "Female Psammomys obesus Are Protected from Circadian Disruption-Induced Glucose Intolerance, Cardiac Fibrosis and Adipocyte Dysfunction"

_ijms, 2024, doi:10.3390/ijms25137265_

Round 1

Reviewer 1 Report

Comments and Suggestions for Authors

The manuscript entitled 'Female Psammomys obesus are protected from circadian disruption-induced glucose intolerance, cardiac fibrosis and adipocyte dysfunction' shows differences between male and female P. obesus with respect to diseases related to circadian disruption. The authors showed that a short photoperiod in males causes glucose intolerance, increased cardiac fibrosis, and increased levels of the inflammatory chemokine Ccl2 in the myocardium. In females, they observed lower levels of Bax protein and markers of cardiac hypertrophy in the hearts, and females were protected from the effects of circadian disruption on obesity-related inflammation and impaired browning.

The authors did a good job of presenting the results. They explained the significance of each of the elements studied in relation to sex and circadian rhythm disruption in the discussion. However, the question arises as to why the authors did not analyze gene expression with ANOVA or the Kruskal-Wallis test but only with the t-test (section 2.3. and Figure 3). Indeed, the differences between male and female NP-M vs. NP-F and SP-M vs. SP-F were presented for all other parameters analyzed, but not here. Is there a specific reason why these analyses were not performed? Could the authors perform these analyses additionally? In this study, it would be interesting to see if there are differences in the expression of the Ccl2, Bax, and Per2 genes between males and females.

The methods are well described, but the manuscript needs to be more consistent regarding the number of animals per group. Namely, in the methods (line 421), it says n=7-9/group, but in most of the figures, it says n=6-9/group, except in Figure 5, where it says n=6/group. This needs to be harmonized in the manuscript or in the methods if a lower number of animals per group (from a minimum of 6) was used.

Reviewer 2 Report

Comments and Suggestions for Authors

The authors have used fat sand rats (Psammomys obesus) which were fed with high fat diet as a model to study the effect of prolonged mild circadian rhythm alteration i.e. 5/19 light/dark instead of 12/12. They find per histology and rtPCR of pro-inflammatory and adipocyte differentiation genes that male rats are more vulnerable to pericardial fibrosis, visceral obesity and pro-inflammatory gene upregulation than female rats.

It has been shown before that male fat sand rats are susceptible to obesity-associated diseases and circadian challenges than female rats. Hence, the study is not entirely novel.

Unfortunately, the authors do not show individual body weights over time which are essential to interpret the histology findings and gene expression and glucose analysis, which is unfortunately only a one-time-point analysis. Time courses along with time courses of body weight would be preferable. The methods state that body weights were monitored before start of circadian challenges. This should be revealed for individual rats. One would also like to see the relative development of body weight (change from baseline) as supplementary information.

Alterations of expression of inflammatory and adipocyte markers are all only at the transcriptional level (rtPCR). For the most prominent and crucial results (crucial for interpretation) one would like to see confirmation at protein level.

The mRNA results of all groups should be normalized to one group and presented all in one figure (e.g. all versus on NP-F) or versus a common mean or median to allow for ANOVA based statistics and assessment of NP vs. SP groups irrespective of sex. All groups should be presented in one figure (i.e. identical Y-axes scaling). Ucp1 mRNA appears not to be normally distributed. Log transformation should be considered.

The histology images are cut outs of whole images which are shown in the supplement. The images in the supplement are much better than the cut-outs and it does not occur why they do not use the images as shown in the supplement (which then would not be needed). Presently they show only one example per group which can hardly be representative of the interindividual variability in such a model that has substantial inherent variability. Therefore, for the most important results e.g. pericardiac fibrosis, cardiac myocyte hypertrophy and visceral fat, the authors should reveal a panel of each three different rats per group (i.e. 4x3) to reveal the variability and provide better insight into histology findings. In addition, it would be helpful to present in a new supplement how the histology was quantified i.e. for example showing binary masks of structures and measurements. Presently, there is no description of the quantification of fibrosis and myocyte hypertrophy. The diameters (or areas?) of fat cells were obtained by manually encircling the cells. This methods appears to be quite limited and it is not told how many of such cells were analysed per section per rat. For example ImageJ provides tools for cell area/volume assessment.

Although the results are presented as scatter plots representing individual rats throughout the figures, the presentations do not allow for assessment how one individual rat responded in different crucial readouts and gene expression, which could be revealed for example by xy-scatter plots with regression analyses. Multiple rtPCR results could be revealed in a heatmap so that one could see each animal for each gene on one-view. One could also depict each scatter of an animal of a group in specific colors (i.e. each animal gets a certain color) or by use of different symbols per animal (for example using letters or numbers as scatter symbols).

In some figures Y-axes labelling do not reveal that results are mRNA (rtPCR) for example Myh ratios. Myh ratios of NP-M and NP-F are missing.

It is likely that owing to the longer dark period rats exposed to the 5/19 rhythm had more time to consume the high fat diet and therefore got fatter and had stronger manifestations of obesity. Hence, one cannot conclude that alterations of circadian rhythms are per se unfavourable. Indeed the one-time-point body weight (table 1) suggest that the group means of the body weights were similar irrespective of the circadian rhythm. This could mean that the longer activity time (dark period) was associated with longer physical activity which offset the unfavourable overeating. The differences between NP and SP would possibly not have occurred if the feeding period had been restricted to e.g. 12 hours irrespective of the circadian settings. Actually it is not told if rats had access to food ad libitum. It needs to be discussed. It is also not told if the histology quantifications and rtPCR analyses were done with or without knowledge of group allocations.

A list of primers is missing.

Supplementary materials: the list headings of supplementary figures is missing.

Round 2

Reviewer 2 Report

Comments and Suggestions for Authors

The authors have adequately addressed my comments. The paper is much improved